# Autogenic Language Embedding for Coherent Point Tracking

## ABSTRACT

Point tracking is a challenging task in computer vision, aiming to establish point-wise correspondence across long video sequences. Recent advancements have primarily focused on temporal modeling techniques to improve local feature similarity, often overlooking the valuable semantic consistency inherent in tracked points. In this paper, we introduce a novel approach leveraging language embeddings to enhance the coherence of frame-wise visual features related to the same object. We recognize that videos typically involve a limited number of objects with specific semantics, allowing us to automatically learn language embeddings. Our proposed method, termed autogenic language embedding for visual feature enhancement, strengthens point correspondence in long-term sequences. Unlike existing visual-language schemes, our approach learns text embeddings from visual features through a dedicated mapping network, enabling seamless adaptation to various tracking tasks without explicit text annotations. Additionally, we introduce a consistency decoder that efficiently integrates text tokens into visual features with minimal computational overhead. Through enhanced visual consistency, our approach significantly improves point tracking trajectories in lengthy videos with substantial appearance variations. Extensive experiments on widely-used point tracking benchmarks demonstrate the superior performance of our method, showcasing notable enhancements compared to trackers relying solely on visual cues.

## CCS CONCEPTS

• **Computing methodologies** → **Tracking**; Matching; *Motion capture*; *Interest point and salient region detections*.

## KEYWORDS

Point Tracking, Language Assisted, Semantic Correspondence, Vision-Language Model

## 1 INTRODUCTION

Point tracking represents a cutting-edge approach in the field of visual tracking, aiming to establish pixel-level correspondences across successive video frames. This task poses significant challenges as it requires an implicit understanding of both the structural and dynamic aspects of the scene to ensure accurate tracking. Point tracking is akin to optical flow [14, 38, 49], yet it extends the scope

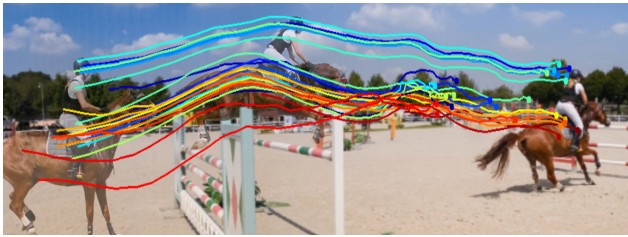

**(a) without autogenic language embedding**

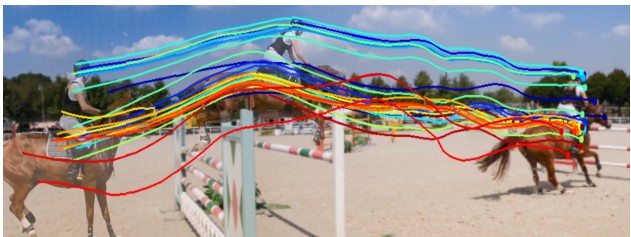

**(b) with autogenic language embedding**

**Figure 1: Visualizing trajectories of tracked points. We visualize the motion of the object at various times and compare the tracking trajectory between the baseline method relying solely on visual features (without autogenic language embedding) and our approach (with autogenic language embedding). Our method maintains the same structural framework as the baseline, differing only in the utilization of language-assisted consistency.**

of point correspondences across substantially longer temporal intervals. Previous research in point tracking has predominantly concentrated on enhancing temporal modeling. This includes strategies such as learning temporal priors for predicting pixel locations [12], identifying robust long-term flow sequences in scenarios involving occlusion [25], and synchronously tracking points across extended frame sequences [16]. These methods primarily leverage similarities in local features across frames, while they are vulnerable to changes in appearance and other variations.

In this paper, we introduce a novel approach by focusing on the semantic coherence of tracked points, a facet that has hitherto been overlooked. We argue that corresponding points across frames should consistently represent the same object and share identical semantics. Typically, the number of objects—and consequently, the semantic groups—in a scene is limited. This observation suggests a straightforward strategy of clustering points into groups and restricting matchings within these clusters. However, such an explicit clustering approach is heavily dependent on the clustering quality and is susceptible to noise. To address these challenges, our work proposes associating features across different frames within a language-assisted semantic space, capitalizing on the expansive and open-ended nature of language semantics. We hypothesize that

the integration of textual tokens into visual features can bridge the spatial discrepancies of identical objects across frames, thereby enhancing semantic consistency. This approach has demonstrated superiority in semantic correspondence tasks when compared to traditional visual hypercolumn features [23].

While it is common to incorporate text embeddings into visual features to condition them [20, 37], applying this technique to point tracking introduces unique challenges. For one, point tracking tasks generally lack associated textual data, making it unfeasible to consistently input accurate descriptions for each video sequence manually. Furthermore, point tracking often relies on lightweight convolutional neural networks to satisfy the constraints of real-time processing and handling multiple frames, presenting a stark contrast to the more complex architectures typically used in vision-language tasks.

To address these challenges, we propose a coherent point tracking framework augmented with **A**utogenic **L**anguage embedding, termed ALTracker. This autogenic language-assisted strategic emphasis ensures that, even in lengthy sequences with substantial appearance variations, as shown in Figure 1, our approach (with autogenic language embedding) demonstrates more robust semantic correspondence compared to approaches relying solely on visual features (without autogenic language embedding). Our approach comprises three key components: 1. an automatic text prompt generation module that generates text tokens from image features through a vision-language mapping network; 2. a text embedding enhancement module, ensuring precise text descriptions by incorporating image embeddings; and 3. a text-image integration module designed to enrich the consistency of image features with textual information. In contrast to other vision-language tasks, our text information is automatically generated from image features, thus it can be adapted to any tracking task without requiring explicit text data. In addition, our visual consistency enhancement approach can be plugged into any point tracking method to effectively improve the tracking performance with slight computation overhead. Applying our feature enhancement to the baseline tracker enhances the Average Jaccard (AJ) score from 54.2 to 61.6 on the TAP-Vid-DAVIS [6] dataset. Extensive comparison experiments on several challenging datasets including TAP-Vid [6] and PointOdyssey [47] exhibit state-of-the-art performance, which further evidences the correctness of our analysis regarding the language impact on visual features.

In summary, our main contributions include:

(1) We demonstrate that text prompts notably strengthen visual consistency, with detailed textual descriptions providing a greater contribution to semantic correspondence. Following this revelation, we utilized this insight for point tracking in long video sequences.

(2) We propose a autogenic language-assisted point tracking approach. Our text embedding is learned from visual features through a specialized mapping network, and we design a consistency decoder which efficiently incorporates text tokens into visual features with minimal computational overhead. As text information is automatically generated from visual features, our approach can be seamlessly adapted to any tracking task without requiring explicit text description.

## 2 RELATED WORK

*Optical flow.* Optical flow aims to attain pixel-level motion estimation of objects in image pairs. Traditionally, optical flow is conceptualized as an optimization problem and addressed through variational methods [2, 3, 14, 17, 19, 49]. Presently, convolutional network-based methods have demonstrated superior performance. FlowNet [9] employs a deep learning framework to learn end-to-end optical flow estimation models. DCFlow [43] constructs a 4D cost volume with convolutional features and refines the cost volume through Semi Global Matching. PWCNet [35] reduces computing costs by employing a feature pyramid to learn multi-scale features and incorporating wrapping techniques. RAFT [38] extracts pixel-level features, generates a 4D cost volume for each pixel, and iteratively updates the optical flow field by searching the cost volume. Recently, transformers [40] have made significant strides in optical flow research. FlowFormer [15] encodes the 4D cost volume into cost memory using an alternative group transformer layer and decodes the location cost queries through a recurrent decoder. GMFlow [46] formulates the flow estimation as a global matching problem, acquiring the matching relationship through a direct comparison of feature similarities. While optical flow methods allow for precise motion estimation between consecutive frames, they are not suited to long-range motion estimation.

*Point Tracking.* Several works develop the point tracker for predicting long-range pixel-level tracks in a feedforward manner. TAP-Vid [6] formulates the problem of Tracking Any Point (TAP) as continuously tracking target points through occlusion in a video sequence, which calculates a cost volume independently for each frame pair and utilizes it for coordinate regression and occlusion branches. Particle Video Revisited (PIPs) [12] revisits the classic Particle Video [31] problem, presenting a model which iteratively refines the features of multiple consecutive frames within a sliding window, enabling the prediction of the tracking point's trajectory and visibility. Recently, many concurrent works have emerged. MFT [25] identifies the most reliable sequence of flows by considering the occlusion and uncertainty map. Context-TAP [1] enhances PIPs by incorporating spatial context during the tracking of trajectories. TAPIR [7] combines two-stage approaches: a matching stage inspired by TAP-Net and a refinement stage inspired by PIPs. PointOdyssey [47] extends PIPs by removing its rigid 8-frame constraint, enabling it to consider a much broader temporal context. OmniMotion [41] represents a video using a quasi-3D canonical volume and achieves pixel-wise tracking through bijections between local and canonical space. CoTracker [16] collectively models the correlation of different points in time through specialized attention layers and iteratively updates the trajectories. Our contribution is complementary to these works: the language information can be automatically generated and embedded into the visual feature to enhance the consistency across long-range video frames.

*Vision-language models.* Recently, the CLIP model [28] measures the similarity between images and text, it maps images and their corresponding text descriptions into a shared feature space that allows the model to perform various tasks, such as image segmentation [29], few-shot learning [39] and image caption [24]. Several methods explore utilizing language signals to the area of object

**Figure 2: Visualization of semantic correspondence with various text prompts. The leftmost image is the source image with a set of key points; target images on the right part show correspondence results under various text prompts. We use circles to denote correctly-predicted points under the threshold $\alpha_{bbox} \leq 0.1$ and crosses for incorrect matches.**

tracking [21, 33, 34, 45]. Some trackers [10, 42, 44] use the language signal as an additional cue and combine it with the commonly used visual cue to compute the final tracking result. SNLT tracker [10] exploits visual and language descriptions individually to predict the target state and then dynamically aggregates these predictions for

generating the final tracking result. Other methods [11, 22] focuses on integrating the visual and textual signals to get an enhanced representation for visual tracking. CapsuleTNL [22] develops a visual-textual routing module and a textual-visual routing module to promote the relationships within the feature embedding space

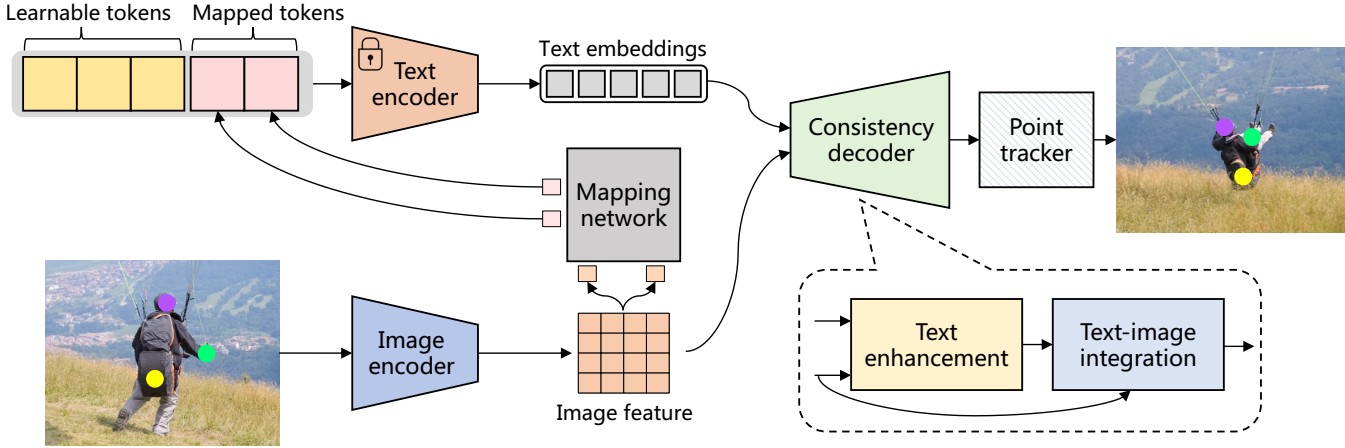

**Figure 3: The architecture of our ALTracker. We introduce a mapping network that aligns image features with corresponding mapped tokens to automatically obtain the text information. A consistency decoder is designed to jointly process textual and visual information, the text enhancement module refines text embedding with enhanced descriptive capabilities, and an image-text integration module integrates the enhanced text embeddings seamlessly into image features. Finally, the tracking result is obtained through any point tracker.**

of query-to-frame and frame-to-query for object tracking. In contrast to previous research, we leverage the semantic information of language to improve consistency in point tracking tasks over long sequences. Furthermore, our approach generates text descriptions from the image example, which eliminates the need for language annotations and expands the range of potential applications.

## 3 METHOD

In this section, we introduce our coherent point tracker approach with autogenic language embedding. Before proceeding, we first present an analysis about text prompts in semantic correspondence.

### 3.1 Revisiting text-embedded visual features in semantic correspondence

Text-embedded visual features [20, 37] have recently demonstrated strong dominance in semantic correspondence tasks. These features are mainly derived from generative diffusion networks [13, 30, 32]. The diffusion feature of a given image is defined as feature maps of intermediate layers at a specific time step during the backward diffusion process. The precise correspondences between two different images can be established using a straightforward approach: locating the maximum cosine similarity of feature maps between the target image and the search image. We refer readers to [37] for more details.

Our analysis highlights the pivotal role played by text prompts in semantic correspondence. Since the diffusion feature [37] used in the semantic correspondence task integrates text embedding as conditions within visual features, we refer to this as the text-embedded visual feature. This feature processes both an image and a text prompt as inputs, maintaining consistency by utilizing the identical text prompt for various images. In pursuit of this understanding, we conduct experiments by adopting various text prompts on target images to find corresponding points, as visualized

in Figure 2. We identify two key patterns: (1) Consistently using the same text prompt between source image and target image improves the correctness of correspondence, as illustrated in the 2nd and 3rd columns in the right part of Figure 2; (2) The accuracy of text description is a crucial factor influencing association ability, as depicted in the 4th column in the right part of Figure 2.

We hypothesize that the text encoder, pretrained by contrast learning [28], generates text embedding containing semantic information similar to visual representations. This ability facilitates consistency across images when aligning these image features within the shared textual semantic space. Furthermore, finer textual descriptions yield more precise semantic information, a quality that distinctly impacts cross-image correspondence.

### 3.2 Autogenic language-assisted tracker

According to the above findings, we propose leveraging the text-embedded visual feature to improve the performance of point tracking algorithms. However, directly incorporating the vision-language model into the point tracking framework poses challenges due to the intricate nature of its feature extraction network. The computational load associated with extracting a single image of the vision-language model is already considerable, making it challenging to fulfill the simultaneous processing demands for long sequences in tracking. Additionally, most datasets in the tracking field lack text information, it is difficult to obtain the detailed description of each sequence through manual input.

We propose the ALTracker, an autogenic language-assisted visual feature enhancement for point tracking to integrate consistency of text prompts into a lightweight image encoder, as illustrated in Figure 3. For text descriptions not available in the tracking dataset, we designed the automatic generation strategy of text tokens, which are generated from learnable tokens and mapped tokens. We introduce a mapping network that aligns image features with corresponding text tokens (i.e. mapped tokens), the text

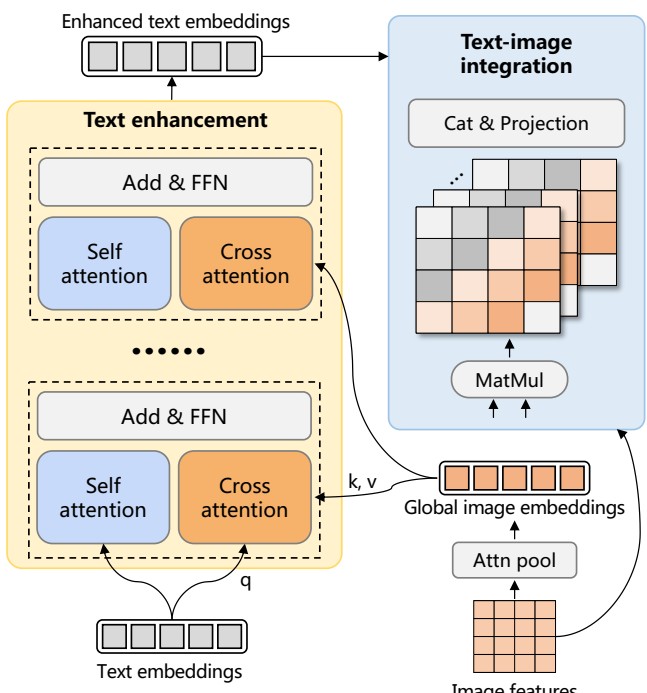

**Figure 4: The architecture of the consistency decoder. Text enhancement module enriches text embeddings by integrating image embeddings into the attention mechanism. Text-image integration module combines enhanced text embeddings with image features to obtain the consistency feature.**

encoder and the image encoder are the same as been adopted in the CLIP [28]. To utilize language consistency in visual features, we incorporate a consistency decoder consisting of a text enhancement module and an image-text integration module. The former refines text embedding with precise descriptive capabilities and the latter integrates the enhanced text embedding into image features.

*3.2.1 Automatic generation of text tokens.* Unlike the original vision-language models that relied on human-designed text prompts, previous language-assisted methods like CoOp [48] and DenseClip [29] introduce learnable text prompts to enhance transferability in downstream tasks. These methods achieved learnable tokens by directly optimizing the text tokens through back-propagation. Taking inspiration from these methods, our framework incorporates the learnable text tokens as a baseline, focusing solely on language-domain prompting. In order to generate visually descriptive text prompts, we introduce mapped tokens on the foundation of learnable tokens as the final text tokens. Mapped tokens are obtained from the image features by a lightweight mapping network. Defining $p$ as the learnable tokens and $x$ as the input image, the final text tokens for the text encoder become:

$$[p, MAP(f_{cls-token}(x))] \qquad (1)$$

where the $f_{cls-token}$ represents the class token of the input image $x$ like the ViT architecture [8]. As the learnable tokens can be easily fine-tuned through the training process, we can employ a simple

Multi-Layer Perception (MLP) as our mapping network *MAP*. The input to the mapping network is the class token, derived from image features via a single-layer attention pooling. This class token encapsulates the global image information and facilitates a more effective alignment with text tokens.

*3.2.2 Vision-language consistency decoder.* The consistency decoder comprises a text enhancement module and an image-text integration module. The text enhancement module refines the text embeddings, endowing them with precise descriptive capabilities, while the image-text integration module seamlessly incorporates the enhanced text embeddings into the image features.

**Text description enhancement** enhances the accuracy of text description by integrating global image embeddings into the text embeddings, which is inspired by the findings that the accuracy of text description is a crucial factor influencing association ability. For example, "*a sitting cat with gray and white color*" is more accurate than "*a cat*" and performs more effectively in semantic correspondence. Based on the basic attention block in the semantic correspondence network [37], we introduce a text enhancement module that incorporates self-attention and cross-attention, as illustrated in Figure 4. Specifically, self-attention operates on the text embedding, while cross-attention operates jointly on the text embedding and the global image embedding, the text embedding serves as the query (q), and the image embedding functions as the key (k) and value (v). Throughout the multi-layer iteration, the text embedding undergoes continuous updates, while the global image embedding is consistently sourced from the original input. The global image embedding is derived by flatting the image feature through an attention pooling layer.

**Image-text integration** is proposed to model the interactions between vision and language to obtain features with semantic consistency. By employing matrix multiplication, we combine enhanced text embedding $t \in \mathbb{R}^{K \times d}$ with image features $x_I \in \mathbb{R}^{H \times W \times d}$ to derive a integrated map $z = x_I t^T, z \in \mathbb{R}^{H \times W \times K}$. This integrated map can be regarded as a mapping result in a consistent space, offering consistent semantic information across a long sequence. The integrated map is concatenated with the image features to yield final features $x_f \in \mathbb{R}^{H \times W \times d}$, as defined by the equation:

$$x_f = Proj(Cat(x_I, x_I t^T)) \qquad (2)$$

where the *Proj* indicates the linear projection and the *Cat* means the concatenation along the last dimension. The obtained final features are used as input to the point tracker to estimate trajectories of interested points. Our framework is scalable and can be applied to many video tasks to enhance feature consistency across long video sequences.

## 3.3 Training Process

We use an off-the-shelf training process for point tracking (e.g., [17, 33, 72]) and freeze the text encoder parameters during training. The other parameters of our ALTracker are optimized by minimizing the sum of a track regression loss which is the L1 distance between the estimated track locations and the ground-truth track locations, and a visibility prediction objective which is the binary cross entropy between the predicted and the ground-truth masks. We employ the point motion estimation in an unrolled window manner as in the

Table 1: Evaluation on TAP-Vid-DAVIS and TAP-Vid-Kinetics datasets [6]. The methods are evaluated under the "queried first" protocol and the "queried strided" protocol on DAVIS. The baseline tracker removing the language-assisted function of our ALTracker and solely rely on visual feature.

| Method | Kinetics First | | | DAVIS First | | | DAVIS Strided | | |
|---|---|---|---|---|---|---|---|---|---|
| | AJ | $< \delta_{avg}^x$ | OA | AJ | $< \delta_{avg}^x$ | OA | AJ | $< \delta_{avg}^x$ | OA |
| TAP-Net [6] | 38.5 | 54.4 | 80.6 | 33.0 | 48.6 | 78.8 | 38.4 | 53.1 | 82.3 |
| PIPs [12] | 31.7 | 53.7 | 72.9 | 42.2 | 64.8 | 77.7 | 52.4 | 70.0 | 83.6 |
| MFT [25] | - | - | - | 47.3 | 66.8 | 77.8 | 56.1 | 70.8 | 86.9 |
| OmniMotion [41] | - | - | - | - | - | - | 51.7 | 67.5 | 85.3 |
| TAPIR [7] | **49.6** | 64.2 | 85.0 | 56.2 | 70.0 | 86.5 | 61.3 | 73.6 | 88.8 |
| CoTracker [16] | 48.7 | **64.3** | **86.5** | 60.6 | 75.4 | **89.3** | 64.8 | **79.1** | 88.7 |
| Baseline | 45.6 | 62.2 | 83.9 | 54.2 | 72.7 | 81.5 | 60.2 | 74.7 | 88.0 |
| ALTracker(ours) | 48.7 | **64.3** | 85.8 | **61.6** | **75.5** | **89.3** | **65.2** | 79.0 | **89.1** |

CoTracker [16]. The primary track regression loss can be defined as:

$$\mathcal{L}_{pri} = \sum_{j=1}^{J} ||\hat{P}^{(m)} - P^{(j)}|| \qquad (3)$$

in this equation, $\hat{P}^{(j)}$ represents the estimated trajectories, while $P^{(j)}$ denotes the ground-truth trajectories, both specific to window j. For trajectories starting in the middle of the window, backwards padding is applied.

## 4 EXPERIMENTS

In this section, we verify the distinct contributions in the ablation study, and present the tracking evaluation on several challenging benchmarks containing manually annotated trajectories in real videos, including PointOdyssey [47], TAP-Vid-DAVIS [6] and TAP-Vid-Kinetics [6].

### 4.1 Setting

**Implementation Details**. We employ a text encoder and an image encoder in the CLIP [28], the point motion estimation in the CoTracker [16]. In the training phase, we train our approach on the PointOdyssey [47] training set for 80,000 iterations, and we randomly choose a 1/2/3-interval for consecutive frames. Points are preferentially sampled on objects and we randomly sample 256 trajectories for each batch, with points visible either in the first or in the middle frame. The size of an input image is resized to 384×512. The AdamW [18] optimizer is employed with an initial learning rate of $5e^{-4}$. We train our model on 4 Nvidia Tesla V100 GPUs. The mini-batch size is set to 4 with each GPU hosting 1 batch. Our approach is implemented in Python 3.8 with PyTorch 1.10.

**Datasets**. PointOdyssey [47] is a large-scale synthetic dataset for long-term point tracking of 80 videos on training set, 11 videos on validation set, and 12 videos on test set, with 2035 average frames and 18,700 tracks per video. TAP-Vid-DAVIS [6] is a real-world dataset of 30 videos from the DAVIS 2017 val set [27], which clips ranging from 34~104 frames and an average of 21.7 point annotations per video. TAP-Vid-Kinetics [6] is a real-world dataset of 1,189 videos each with 250 frames from the Kinetics-700-2020 val set [5] with an average of 26.3 point annotations per video.

**Evaluation Metrics**. We report both the position and occlusion accuracy of predicted tracks. Following the TAP-Vid and PointOdyssy benchmarks, our evaluation metrics include: Average Position Accuracy ($< \delta_{avg}^x$) measures the average position accuracy of visible points over 5 threholds {1, 2, 4, 8, 16}; Average Jaccard (AJ) evaluates both occlusion and position accuracy on the same thresholds as $< \delta_{avg}^x$; Occlusion Accuracy (OA) evaluates the accuracy of the visibility/occlusion prediction at each frame; Median Trajectory Error (MTE) measures the distance between the estimated tracks and ground truth tracks; "Survival" rate means the average number of frames until tracking failure and is reported as a ratio of video length, failure is when L2 distance exceeds 50 pixels.

Table 2: Evaluation on PointOdyssey test set [47].

| Method | MTE↓ | $\delta$ ↑ | Survival↑ |
|---|---|---|---|
| RAFT [38] | 319.46 | 23.75 | 17.01 |
| DINO [4] | 118.38 | 10.07 | 32.61 |
| TAP-Net [6] | 63.51 | 28.37 | 18.27 |
| PIPs [12] | 63.98 | 27.34 | 42.33 |
| PIPs++ [47] | 26.95 | 33.64 | 50.47 |
| Baseline | 27.53 | 29.41 | 49.22 |
| ALTracker(ours) | **24.44** | **33.91** | **51.37** |

### 4.2 State-of-the-art Comparison

We set the tracker solely rely on visual features as our baseline, and compare the performance of a autogenic language-assisted visual tracker with the baseline tracker. The baseline tracker uses the same image encoder and motion estimation modules as ALTracker, removing the language-assisted function including text encoder, mapping network and consistency decoder.

**TAP-Vid**. For the TAP-Vid benchmarks, we follow the standard protocol and downsample videos to 256 × 256 before passing them to the model, all the metrics are then computed in 256 × 256. We evaluate our models on the TAP-Vid-DAVIS and TAP-Vid-Kinetics, points are queried on objects at random frames and the goal is to predict positions and occlusion labels of queried points. In the

**Table 3: Ablation experiments on the text generation and text enhancement module. In the text generation module, we present the effectiveness of learning tokens and the type of mapping network. In the text enhancement module, we provide the evaluation of self-attention layers (self) and cross-attention layers (cross). Default settings are marked in gray.**

| Learnable tokens | Mapping network | Text enhancement | | Kinetics First | | | DAVIS First | | |
|---|---|---|---|---|---|---|---|---|---|
| | | self | cross | AJ | $< \delta_{avg}^x$ | OA | AJ | $< \delta_{avg}^x$ | OA |
| - | MLP | - | - | 40.6 | 55.2 | 76.7 | 42.2 | 61.2 | 72.3 |
| ✓ | MLP | 6 | - | 45.6 | 61.2 | 81.1 | 58.8 | 73.5 | 85.2 |
| ✓ | MLP | - | 6 | 44.3 | 60.1 | 80.0 | 53.5 | 68.7 | 83.9 |
| ✓ | MLP | 4 | 4 | 46.6 | 60.7 | 84.9 | 60.8 | 75.1 | 89.1 |
| ✓ | MLP | 6 | 6 | 48.7 | 64.3 | 85.8 | 61.6 | 75.5 | 89.3 |
| - | MLP | 6 | 6 | 48.0 | 63.2 | 85.8 | 59.5 | 74.6 | 88.6 |
| ✓ | Transformer | 6 | 6 | 46.2 | 62.9 | 83.6 | 60.1 | 75.1 | 87.2 |
| ✓ | MLP | 10 | 10 | 48.7 | 64.3 | 85.8 | 61.6 | 75.5 | 89.4 |

TAP-Vid, "queried first" evaluation protocol, each point is queried only once in the video, at the first frame where it becomes visible. Hence, the model should predict positions only for future frames. In the "queried strided" protocol, points are queried every five frames and tracking should be done in both directions. We adopt the online method as our point tracker, it tracks points only forward, and we run the tracker forward and backward starting from each queried point. As "queried first" requires estimating the longest tracks, it is a more difficult setting than "strided". Moreover, "strided" demands estimating the same track from multiple starting locations and is thus much more computationally expensive. From the experiment results in Table 1, we can see that our method has achieved remarkable performance with an AJ score of 48.7 in Kinetics First and 61.6 in DAVIS First. Furthermore, our method has made significant improvements in all evaluation metrics compared to the baseline.

**PointOdyssey**. Inspecting results (as shown in Table 2) across rows, we can see that our ALTracker achieves the best results among all methods, achieving the highest MTE score of 24.4. Especially, compared to the baseline, our method has achieved a significant improvement by a specific gain of 3.09 of MTE. Our approach significantly outperforms the best existing tracker and demonstrates the effectiveness of the language-assisted feature consistency.

## 4.3    Ablation Study

We ablate our approach to verify the effectiveness of our design decisions using the TAP-Vid-DAVIS and TAP-Vid-Kinetics datasets.

**Automatic generation of text tokens.** We compare the effectiveness of learnable tokens and mapped tokens obtained through different types of mapping network. Learning tokens are plugged into the generated text tokens and can be finetuned through the network training process. From Table 3 we can find that adopting the learning tokens can greatly improve the performance of point tracking (line 5 *vs* line 6 in Table 3). We also conduct the comparison between the MLP and the Transformer as our mapping network, each network adopts a three-layer basic unit. Evaluation results (line 5 *vs* line 7 in Table 3) demonstrate that MLP is a more suitable mapping network than Transformer for our purposes due to its superior performance and lower computational complexity.

**Attention layers in text enhancement module.** We tested several text enhancement schemes, including no text enhancement (line 1 in Table 3), self-attention enhancement only (line 2), cross-attention enhancement only (line 3), and simultaneous self-attention and cross-attention enhancement at different layers (line 4,5,8). Our experimental results demonstrate that both self-attention and cross-attention can enhance the representation ability of text embedding to varying degrees, and the impact remains constant after 6 layers. For optimal accuracy and efficiency, we have chosen 6 layers in our text enhancement module.

**Table 4: Comparison of integration strategies. *Cat* indicates the concatenation, *Map* means the integrated map obtained by the matrix multiplication.**

| Integration | | DAVIS First | | |
|---|---|---|---|---|
| Cat | Map | AJ | $< \delta_{avg}^x$ | OA |
| ✓ | - | 59.1 | 73.6 | 89.5 |
| - | ✓ | 55.9 | 71.0 | 84.7 |
| ✓ | ✓ | 61.6 | 75.5 | 89.3 |

**Text-image integration methods.** Different integration strategies of text embeddings and image features have a large effect on the performance of consistency. A simple way to approach this is to concatenate these two cues (Cat) by flatting the image features and finally map them back to the original size. Our proposed integrated map (Map) through the matrix multiplication of text and image embeddings, which can be used as features alone (line 2 in Table 4) or concatenated with the original image features (line 3) for tracking purposes. The comparison results show that the concatenation of image features and integrated map can effectively improve the tracking performance.

## 4.4    Visualization

We offer visualizations of long-range trajectories of prototypical challenging scenarios to demonstrate the tracking performance of the proposed autogenic language-assisted tracker. Figure 5 shows queried points of the shown frame between our method which adopts the language-assisted consistency (*w* language embedding)

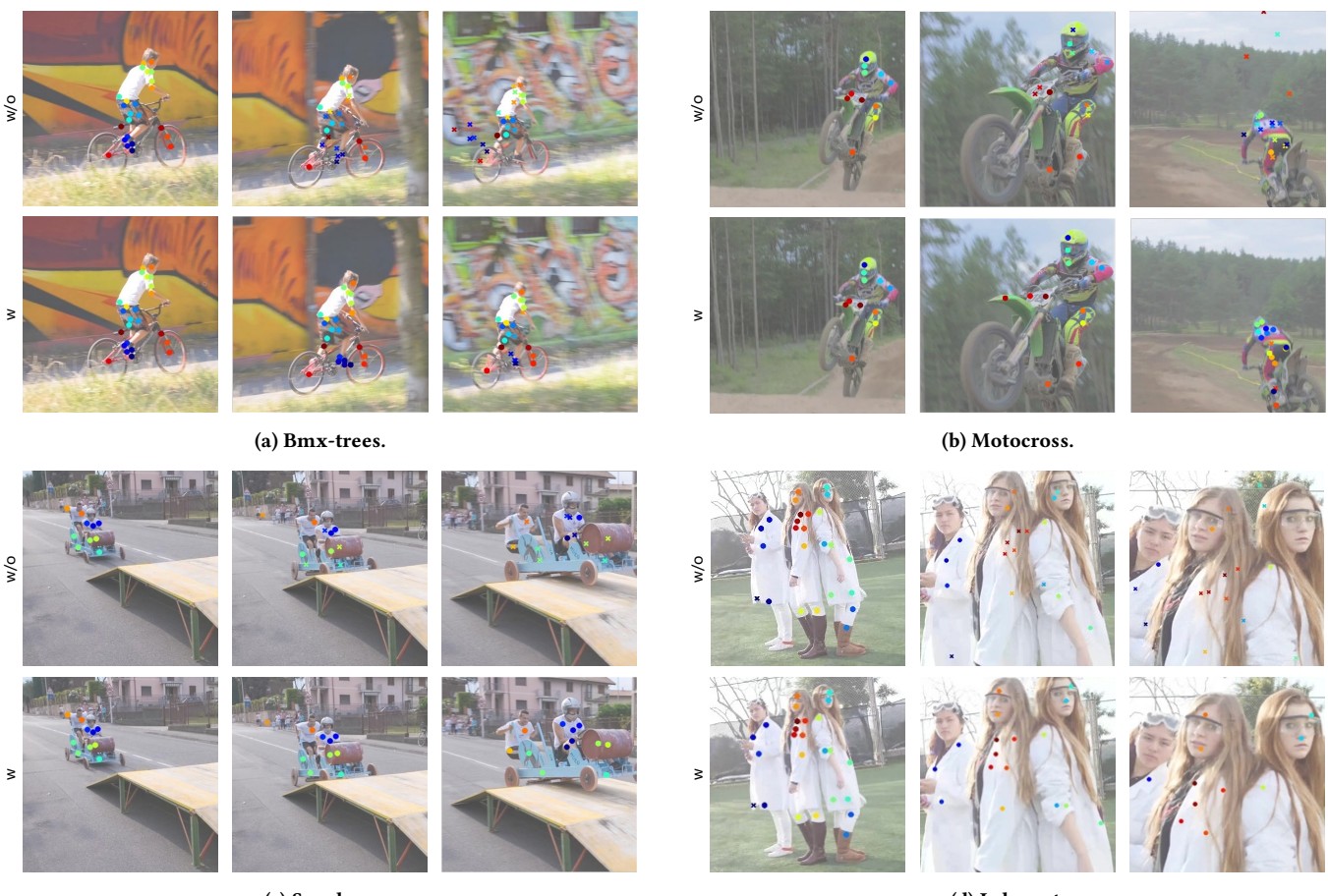

(a) Bmx-trees.

(b) Motocross.

(c) Soapbox.

(d) Lab-coat.

**Figure 5: Visualization of point trajectories on DAVIS [27]. We compare the visualization result between our ALTracker with autogenic language embedding (*w*) and the baseline tracker without language information (*w/o*). The images show tracking results over time. Different colors indicate different points. We use circles to indicate correctly-predicted points under the threshold $\alpha_{bbox} \leq 0.1$ and crosses for incorrect matches. Notably, our method yields accurate, coherent long-range motion even for fast moving (*Bmx-trees*), object deformation (*Motocross*), scale change (*Soapbox*), and similar distractor (*Lab-coat*) scenarios.**

and baseline tracker solely rely on the visual feature (*w/o* language embedding). As can be seen from the figure, after a long period of tracking, our ALTracker still achieves correct point tracking results. Crosses in images indicate incorrect matches. We observe that our approach has a strong discriminative ability for targets with severe scale variations and keeps a reliable associative ability in many challenging scenes.

# 5 CONCLUSION

In this study, we conduct an analysis to elucidate the factors contributing to the robust semantic correspondence. And reveal that text prompts significantly enhance visual correspondence across visual semantics, and precise textual descriptions contribute to improved semantic consistency. Incorporating this insight into the point tracking task, we propose a coherent point tracking approach with autogenic language embedding. Our ALTracker consists of an automatic generation module which adopts a mapping network to

align image features with corresponding text tokens, and a consistency decoder to enhance the descriptive accuracy of text embedding, and then integrate textual information and visual information. As our text information is automatically generated from visual features, it can be seamlessly adapted to any tracking task without requiring for explicit text input. Experiments conducted on several challenging point tracking datasets exhibited impressive performance, demonstrating that our approach renders point tracking more stable and discriminative, particularly in lengthy videos with substantial appearance variations.

**Limitations**. Due to our utilization of the CLIP encoder, our main emphasis was on incorporating autogenic language embedding into convolutional networks for coherent tracking. Nevertheless, we have not yet investigated the potential improvements on alternative visual encoders, such as the various transformers [8, 26, 36]. In the future, we intend to integrate our autogenic language-assisted consistency into more image encoders, enhancing its adaptability to a broader range of scenarios.

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
