# OpenReview forum: "Autogenic Language Embedding for Coherent Point Tracking"
_acmmm.org/ACMMM/2024/Conference — MM2024 Poster_

### Official Review · Reviewer_M1ma · 2024-05-09

**Rating:** 3
**Confidence:** 3

**Summary:**

The paper is about a coherent point tracking framework augmented with Autogenic Language embedding, termed ALTracker. The authors propose a method that integrates text information into visual features to improve the semantic coherence of tracked points in point tracking tasks. They introduce a mapping network that aligns image features with corresponding text tokens, a text embedding enhancement module to ensure precise text descriptions, and a text-image integration module to enrich the consistency of image features with textual information. The proposed approach demonstrates superior performance in semantic correspondence and point tracking tasks compared to traditional visual features.

**Strengths:**

Novelty: The paper introduces a novel approach to point tracking by focusing on the semantic coherence of tracked points, which has been overlooked in previous research. The integration of textual tokens into visual features to bridge spatial discrepancies and enhance semantic consistency is a unique contribution.

Technical Correctness: The paper presents a well-designed architecture, consisting of a mapping network, a consistency decoder, a text enhancement module, and an image-text integration module. The integration of these components seamlessly enhances the descriptive capabilities of text embeddings and integrates them into image features. The technical details are well-explained and supported by empirical evidence.

Adequate Evaluation: The paper conducts comprehensive evaluations to demonstrate the effectiveness of the proposed approach. It compares the performance of the ALTracker with baseline methods and existing trackers, showing superior results in terms of semantic correspondence and point tracking. The evaluations are performed on various datasets, ensuring the generalizability of the proposed approach.

Clarity: The paper is well-written and organized, making it easy to understand the motivations, methods, and results. The authors provide clear explanations of the technical concepts and algorithms used, enabling readers to grasp the key ideas easily.

**Limitations:**

1. The authors claim that "We recognize that videos typically involve a limited number of objects with specific semantics, allowing us to automatically learn language embeddings", however, in real-time tracking, the hypothesis of "limited number of objects" is incorrect.
2. In Figure 3, there are two output tokens from the Image feature, while in section 3.2.1, the authors said they only use the CLS token.
3. Still in section 3.2.1, why "the learnable tokens can be easily fine-tuned through the training process"? Is there a corresponding loss function?
4. In section 3.2.2, "Text description enhancement enhances the accuracy of text description by integrating global image embeddings into the text embeddings", why this works?
5. The authors did not mention the FPS, number of parameters and FLOPs, which is important for real-time tracking.

**Suitability:**

2

---

### Official Review · Reviewer_UY3F · 2024-05-27

**Rating:** 3
**Confidence:** 3

**Summary:**

This work aims to address the challenging point tracking problem. The paper proposes autogenic language embedding for visual feature enhancement, strengthens point correspondence in long-term sequences. The text embedding is learned from visual features through an MLP network. Besides, a consistency decoder is designed to incorporate text tokens into visual features with minimal computational overhead.

**Strengths:**

1. The writing of this paper is overall clear.
2. Both quantitative and qualitative experimental results are provided to demonstrate the effectiveness of the proposed method.

**Limitations:**

1. My main concern is the novelty of the paper. The mapping network designed in the paper is just an MLP layer, and the consistency decoder is a conventional cross-modal attention operation.  For the key insight in the paper, autogenic language embedding, I wonder whether it is technical sound. Can you visualize the generated text, or how can you prove that the generated text information is discriminative?
2. The whole framework seems to be a little complex. The reviewer wonders about the running speed and parameters of the proposed method. Is it slower than the SOTA methods?

**Suitability:**

2

---

### Official Review · Reviewer_TTYr · 2024-06-08

**Rating:** 5
**Confidence:** 2

**Summary:**

The paper presents a novel approach for point tracking aimed at utilising text-based information to enhance the tracking coherence.
The main idea behind this work is to leverage the semantic coherence of the to-be-tracked points, and associate the similar features across different frames using the semantic space represented by language.

First, the authors present a mapping network for the automatic generation of text-tokens.
Second, they also present a visual-language consistency decoder, composed of a text-description enhancer module and an image-text integration module.
Finally, they explicit the training procedure and show the performances of the proposed method on a variety of point-tracking datasets.

**Strengths:**

The paper proposes an interesting and original approach that produces well-performing results.
The said results seems to indicate good performances on a variety of point-tracking datasets, consistent and on par or event superior to state-of-the-art methods.

The overall architecture is simple and lightweight, promising efficiency at inference time, necessary for real-time point tracking.

A significant of ablation studies are present and useful to clarify doubts about possible implementation design and to delve deeper into the proposed method.

**Limitations:**

The proposed approach seems to heavily rely on the goodness of the text-encoder performances, with my major doubt regarding its usage and performances on scenes with an high-degree of semantic complexity with varying and diverse objects between frames and their impact on the text embedding.

The paper, while mostly well written, still presents some unclear sections with evident repetitions (Eg. lines 499-501 and lines 531-534).
I believe that the paper can still be improved when regarding its overall readability.

**Suitability:**

3

---

### Meta-Review · Area_Chair_nmLK · 2024-07-10

**Recommendation:** Accept (Poster)
**Confidence:** 4

**Metareview:**

This paper shows how to leverage textual descriptions to improve point tracking. The approach is novel and interesting and experiments are carried thoroughly. The multimodality of the approach make it a great fit for MM. All in all positive comments overcome the negative review.